# Genotyping of African Swine Fever Virus (ASFV) Isolates in Romania with the First Report of Genotype II in Symptomatic Pigs

**DOI:** 10.3390/vetsci8120290

**Published:** 2021-11-26

**Authors:** Andrei Ungur, Cristina Daniela Cazan, Luciana Cătălina Panait, Marian Taulescu, Oana Maria Balmoș, Marian Mihaiu, Florica Bărbuceanu, Andrei Daniel Mihalca, Cornel Cătoi

**Affiliations:** 1Department of Pathology, Faculty of Veterinary Medicine, University of Agricultural Sciences and Veterinary Medicine of Cluj-Napoca, Calea Mănăștur 3-5, 400372 Cluj-Napoca, Romania; andrei.ungur@usamvcluj.ro (A.U.); marian.taulescu@usamvcluj.ro (M.T.); cornel.catoi@usamvcluj.ro (C.C.); 2Molecular Biology and Veterinary Parasitology Unit (CDS-9), Faculty of Veterinary Medicine, University of Agricultural Sciences and Veterinary Medicine of Cluj-Napoca, Calea Mănăștur 3-5, 400372 Cluj-Napoca, Romania; 3Department of Parasitology and Parasitic Diseases, Faculty of Veterinary Medicine, University of Agricultural Sciences and Veterinary Medicine of Cluj-Napoca, Calea Mănăștur 3-5, 400372 Cluj-Napoca, Romania; luciana.rus@usamvcluj.ro (L.C.P.); oana-maria.balmos@usamvcluj.ro (O.M.B.); amihalca@usamvcluj.ro (A.D.M.); 4Department of Food Safety, Faculty of Veterinary Medicine, University of Agricultural Sciences and Veterinary Medicine of Cluj-Napoca, Calea Mănăștur 3-5, 400372 Cluj-Napoca, Romania; mihaiu.marian@usamvcluj.ro; 5Institute for Diagnosis and Animal Health, 050556 Bucharest, Romania; florica.barbuceanu@idah.ro; 6Technological Transfer Centre for Animal Nutrition and Comparative Pathology ‘COMPAC’ of the University of Agricultural Sciences and Veterinary Medicine of Cluj-Napoca, Calea Mănăștur 3-5, 400372 Cluj-Napoca, Romania

**Keywords:** African swine fever virus (ASFV), domestic pigs, genotype II, p72, Romania

## Abstract

The World Organisation for Animal Health has listed African swine fever as the most important deadly disease in domestic swine around the world. The virus was recently brought from South-East Africa to Georgia in 2007, and it has since expanded to Russia, Eastern Europe, China, and Southeast Asia, having a devastating impact on the global swine industry and economy. In this study, we report for the first time the molecular characterization of nine African swine fever virus (ASFV) isolates obtained from domestic pigs in Mureş County, Romania. All nine Romanian samples clustered within p72 genotype II and showed 100% identity with all compared isolates from Georgia, Armenia, Russia, Azerbaijan, Ukraine, Belarus, Lithuania, and Poland. This is the first report of ASFV genotype II in the country.

## 1. Introduction

The World Organisation for Animal Health (OIE) considers African swine fever (ASF) one of the most severe and important disease affecting the domestic pig and wild boar populations around the world [1]. The high mortality rates among domestic pigs are also devastating to the swine industry, with severe effects on the economy [2]. The high economic impact in the swine industry is currently influenced by the lack of specific medical therapy [2]. No vaccine is currently available against the disease [2]. As no medical technologies have been developed yet against the disease, the only way to slow down the spreading of its etiological agent, African swine fever virus (ASFV), is to quarantine all affected regions. The swine populations that are positive to ASF are currently culled, and the carcasses are destroyed by incineration. In some circumstances, a prophylactic depopulation is also performed throughout the positive regions. Furthermore, the commercialization and movement of domestic pigs and pig products are prohibited. As a result, not only the lethality of ASF but also the control efforts inflict significant economic losses to all farms from all positive countries [3]. Considering that wild boar populations can be affected by ASFV in the same ways as domestic pigs, the control and eradication of the disease is considerably more challenging in the affected countries due to the free movement of wild boars and their active contaminant role [3].

ASF is a viral disease widespread in Africa, Asia, and Europe, with acute-to-chronic manifestation and a hemorrhagic character. Recent studies have disputed the highly contagious character and proved that ASF spreads rather slowly compared with other infectious diseases [4]. Both domestic pigs and wild boar populations are the main species affected. The disease is also characterized by severe hyperthermia, abortion, hyperemic areas in the skin, and hemorrhages in several internal organs. Several factors, such as the hosts’ immune system status, route of infection, virulence, and dosage of virus, can severely influence the clinical course and the pathological features of the disease. The most virulent strains of ASFV give the most fulminant clinical form, peracute ASF, in which sudden death is noticed [5]. Severe and diffuse splenic enlargement with dark-black discoloration and increased friability of the parenchyma are the only gross features of this clinical form. In the acute form of ASF, the mortality can reach up to 100% during the first week. During the first week of ASFV evolution to a host, the predominant clinical manifestations include emesis, nasal discharge, bloody diarrhea, apathy, abortion, and cutaneous hyperemia. This clinical form tends to be the most common form of the disease, with the isolates being moderately to highly virulent. The postmortem findings include the ones from the peracute form, followed by hemorrhagic necrotizing lymphadenitis. In some situations, petechiae might be observed in several organs such as the heart, kidneys, and urinary bladder [6]. When the viral strains are moderately virulent, similar but less severe clinical signs occur, represented by the subacute form of ASF, and the gross findings are more intense than in the acute form. The chronic forms of ASF are caused by low-to-moderate virulence strains. The clinical signs include multifocal necrosis in the skin, intermittent fever, respiratory distress, arthritis, and loss of weight [5], and the gross features are represented by glomerulonephritis, chronic pericarditis, pneumonia, skin lesions, fibrinous arthritis, and pleural adhesions [7].

In the European continent, two circulating genotypes of ASFV were previously identified, each of them having a different place of origin. The genotype I incursion started in Portugal in 1960, spreading to Spain and several other European countries, and it was eradicated by 1995, with the exception of Sardinia, Italy, where it remained endemic [8]. ASFV genotype II was firstly reported in Georgia in 2007 [9]; since then, it has spread to countries in eastern and central Europe [1]. A recent study confirmed that the circulating strains of ASFV in Asia are derived from the ones in Europe and Russia [10]. ASF was first reported in the European Union in Lithuania in early 2014, and it has since spread to several European Union countries, including Romania [11]. An important factor in the epidemiology of ASFV are the wild boar populations [12]. This specific group plays a major role in the dissemination of ASFV in most affected countries, as a natural epidemiological reservoir, allowing the virus to persist in the environment indefinitely without the presence of any other susceptible animals or vectors [12]. The first confirmation of ASFV in Romania was in the summer of 2017, in a backyard holding in the north-western region of the country [13]. From the first outbreak, the situation rapidly escalated, and infections with ASFV were confirmed in all Romanian counties, in both domestic pigs and wild boars. The rapid, extensive spread of the virus in Romania may have been influenced by the traditional extensive rearing system represented by backyard farms, and its feeding, watering, and housing conditions [14].

ASFV is a member of the *Asfivirus* genus from the Asfarviridae family. It is a complex enveloped virus with an icosahedral morphology and a double-stranded DNA [15]. The main site of replication is the cytoplasm of the infected cells, although an earlier stage of nuclear DNA synthesis has been documented [16,17], DNA replication occurs in the perinuclear factory site about 6 h after infection. Enzymes essential for virus genome transcription and replication, as well as virion structural proteins, are encoded in the virus genome [18]. The genome sequence size varies between 170 kilo base pairs (kbp) and 194 kbp long, and it is encoding around 150 to 167 open reading frames [19]. There are three main areas of the genome. The core region has around 125 kbp and is always constant in its length, with size variations lower than 1.5%. The core area is flanked on both ends by two highly variable regions [20]. There are zones of localized high variability within the central constant region, such as the CVR area within the B602L gene, caused by differences in the number of tandem repeats, some of which are utilized to distinguish different isolates [21]. Based on sequence differences in the C-terminal region of the B646L gene, which encodes the primary variable capsid protein p72, ASFV strains and isolates were classified into 24 genotypes, reported in eastern and southern Africa, with genotypes I and II having become established in other regions as well. To follow virus evolution across countries and regions, information based solely on the B646L can be used for an initial sequencing of the partial genome, but for a full genome sequence, the usage of sub-groupings based on intergenic sections is necessary [22].

From the first report of an ASF case in 2017, there were more than 3800 confirmed ASF outbreaks in Romania, with a countrywide distribution [23]. However, no data regarding the circulating genotypes are currently available in the country. The present study aimed to identify the circulating ASFV genotype in a county from the central part of Romania in the context of ASF outbreak.

## 2. Materials and Methods

In order to assess the ASFV genotype present in Romania, 9 previously extracted DNA blood samples from Veterinary Health and Food Safety Department (DSVSA) Cluj were randomly selected and molecularly investigated. The selected samples originated from Mureş County (central region of Romania). Sample details are shown in Table 1.

For the p72 genotype classification, the C-terminal region of the p72 protein gene (B646L) was amplified by conventional polymerase chain reaction (PCR). PCR amplification of the B646L gene region (~478 bp) was performed in a total volume of 25 μL, containing 12.5 μL Red PCR Mastermix (Bioline Meridian Bioscience, Luckenwalde, Germany), 6.5 μL of ultrapure water, 1 μL (10 pmol/μL) of each of the two previously described primers p72-D (5′-GGCACAAGTTCGGACATGT-3′) and p72-U (5′-TACTGTAACGCAGCACAG-3′) [24] and 4 μL aliquot of the previously isolated DNA. One negative control containing 4 μL ultra-pure water instead of DNA was included. The PCR was performed using the T1000™ Thermal Cycler (Bio-Rad, London, UK). The reaction was performed with the following conditions: initial denaturation at 95 °C for 1 min, followed by 35 cycles of denaturation at 95 °C for 15 s, annealing at 54 °C for 15 s and extension at 72 °C for 10 s, with a final extension at 72 °C for 5 s. Electrophoresis was used to visualize the amplification products on 1.5% agarose gel stained with ECO Safe 20,000× Nucleic Acid Staining Solution (Pacific Image Electronics, New Taipei, Taiwan). The molecular weight of the samples was assessed by comparison to a molecular marker (O’GeneRuler™ 100 bp DNA Ladder, Thermo Fisher Scientific Inc., Waltham, MA, USA). All PCR products previously visualized were purified using the ISOLATE II PCR and Gel Kit (Bioline Meridian Bioscience, Luckenwalde, Germany). The sequencing was performed by a private company (Macrogen Europe, Amsterdam, the Netherlands). The obtained sequences were compared with those available in GenBank™ using Basic Local Alignments Tool (BLAST) analyses. All sequences were analyzed and edited using Geneious^®^ 4.85 software [25].

To investigate the phylogenetic analyses among ASFV genotypes, a phylogenetic tree of the B646L gene region was constructed based on all unique sequences obtained in the current study and those available in GenBank. The ASFV genotype II sequences selected from GenBank originated in Georgia (JX857509), Armenia (JX857508), Azerbaijan (JX857515), Belarus (KJ627215), Russia (JX857510), Lithuania (KJ627216), Poland (KJ627218) and Ukraine (JX857521). One B646L sequence of ASFV genotype IX from Congo (HQ645947) was used as an outgroup. The evolutionary history was inferred by using the maximum likelihood method and Jukes–Cantor model [26]. The tree with the highest log likelihood (−586.90) is shown. The percentage of trees in which the associated taxa clustered together is shown next to the branches. Initial tree(s) for the heuristic search were obtained automatically by applying Neighbor-Join and BioNJ algorithms to a matrix of pairwise distances estimated using the Jukes–Cantor model and then selecting the topology with superior log likelihood value. The tree is drawn to scale, with branch lengths measured in the number of substitutions per site. This analysis involved 19 nucleotide sequences. Codon positions included were 1st+2nd+3rd+Noncoding. There was a total of 404 positions in the final dataset. Evolutionary analyses were conducted in MEGA X [27].

## 3. Results

The presence of the viral DNA was confirmed in all nine tested samples. All positive samples were successfully sequenced and phylogenetically analyzed for p72 (B646L) targeted gene. The ASFV Romanian samples clustered within p72 genotype II and showed 100% identity with all compared ASFV isolates from Georgia (JX857509), Armenia (JX857508), Azerbaijan (JX857515), Belarus (KJ627215), Russia (JX857510), Lithuania (KJ627216), Poland (KJ627218), and Ukraine (JX857521) (Figure 1). This is the first report of ASFV genotype II in Romania. The nine Romanian sequences were deposited in GenBank database under the accession numbers OK623917–OK623925.

## 4. Discussion

In the current study, we reported the first partial genome sequence of the circulating ASFV strain in the central region of Romania. As the number of ASF outbreaks in Romania and adjacent countries in Europe grows, scientists are finding it more difficult to pinpoint the spatial and temporal routes of ASFVs producing outbreaks as all ASFVs belong to genotype II. A growing number of complete genomes ASFV sequences obtained from countries around Romania are now available in GenBank, such as the ones from Russia (KP843857), Georgia (FR682468), or Poland (MH681419). Genotype II p72 group ASFV represents by far the most geographically widespread of the 24 viral genotypes currently known, owing to its recent introduction to Georgia and rapid subsequent dissemination north to the Russian Federation, west as far as east Germany, then east to China and other regions in southeastern Asia [28].

Recent studies conducted in China have confirmed that the ASFV strain circulating in the country was derived from an introduction of ASFV strains circulating in Europe and Russia, showing nucleotide identities of 99.944%, 99.951%, and 99.972% with strains from Russia (KP843857), Georgia (FR682468), and Poland (MH681419) [10].

Our genomic data were compared to the most similar partial genome sequences in GenBank [21]; however, none of them differed genetically from the others. All sequenced from the present study showed 100% similarity with all compared ASFV isolates from the previously mentioned countries [21]. All the compared isolates have the same patterns for specific variable regions and cannot be separated into subgroups, suggesting that all the ASF outbreaks in both domestic and wild pigs from Eastern Europe might have the same starting point. The geographical spread of the disease and its distribution can be followed using the timeline of all reported outbreaks in this region, considering that ASFV is very stable and has a long resistance period in both death carcasses and the environment [29]. However, the genotype and subtype analyses of ASFV do not offer enough information regarding the specific source of infection in outbreaks. Other data are necessary, such as historical data and anamnesis in each farm, in order to establish the link between the outbreaks and the possible source of infections [30,31].

## 5. Conclusions

Genetic typing, nucleotide sequencing, and subsequent phylogenetic studies are used to confirm epidemiological data on a qualitatively different level all over the world. It also allows to track the evolution of viruses throughout time. The additional genomic information presented in this work adds to our understanding of the ASFV’s geographical expansion and biological evolution, which can help in developing disease prevention and control techniques. This is the first study evaluating the ASFV genotypes in Romania and the first report of genotype II circulating in the central region of the country. However, more phylogenomic work is needed as the present data could not be generalized to the entire Romanian territory.

## Figures and Tables

**Figure 1 vetsci-08-00290-f001:**
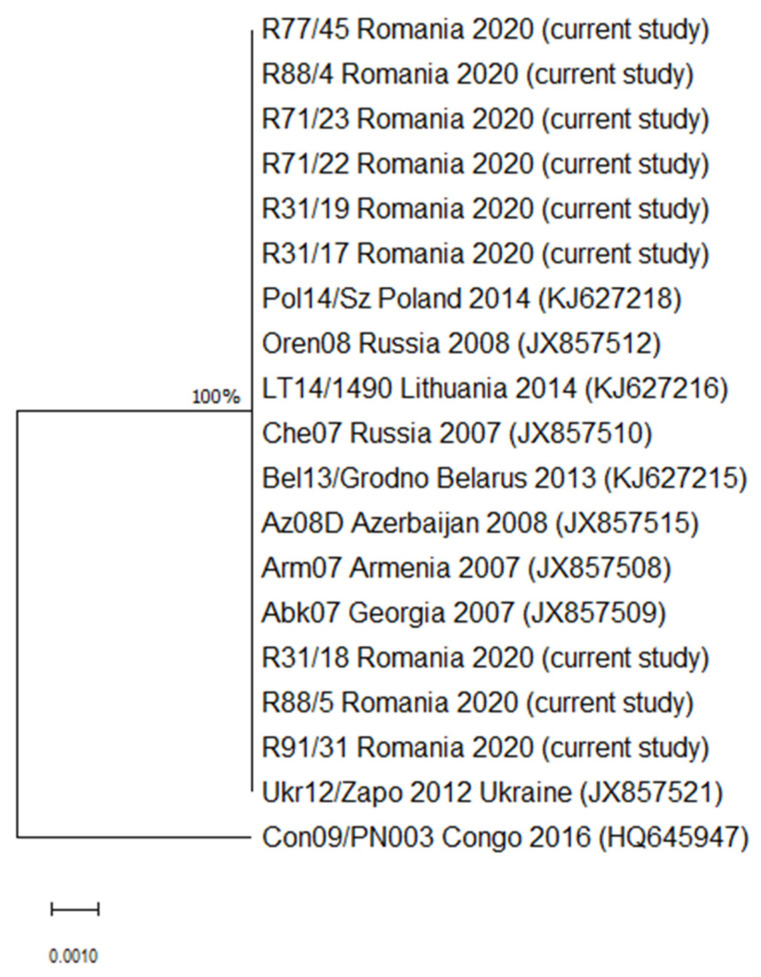
Maximum likelihood tree based on partial p72 B646L rDNA sequences obtained in the current study and sequences of ASFV genotype II (GenBank). Only bootstrap values above 75% are displayed. The scale bar indicates the number of nucleotide substitutions per site. The Gen-Bank accession number, assigned name, and country of origin are indicated for each sequence, if available.

**Table 1 vetsci-08-00290-t001:** African swine fever virus isolates selected for genotyping from wild boars and domestic pigs after virus isolation, Mureș County, Romania, 2020.

Isolate	Host	Locality	p72 Genotype	GenBank Accession Number
R31/17/2020	wild boar	hunting domain 63, Târnăveni	II	OK623917
R31/18/2020	wild boar	hunting domain 63, Târnăveni	II	OK623918
R31/19/2020	wild boar	hunting domain 63, Târnăveni	II	OK623919
R71/22/2020	domestic pig	Dâmbău	II	OK623920
R71/23/2020	domestic pig	Dâmbău	II	OK623921
R77/45/2020	domestic pig	Găneşti	II	OK623922
R88/4/2020	domestic pig	Iernut	II	OK623923
R88/5/2020	domestic pig	Iernut	II	OK623924
R91/31/2020	wild boar	hunting domain 7, Papiu	II	OK623925

## Data Availability

All data generated or analyzed during this study are included in this article.

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
