# Peer review of "Genotyping of African Swine Fever Virus (ASFV) Isolates in Romania with the First Report of Genotype II in Symptomatic Pigs"

_vetsci, 2021, doi:10.3390/vetsci8120290_

Round 1

Reviewer 1 Report

The manuscript submitted by Ungur et al., entitled "Genotyping of African swine fever virus (ASFV) isolates in Romania with the first report of genotype II in symptomatic pigs" aims to characterize nine African swine fever virus (ASFV) isolates obtained from domestic pigs in MureÅŸ County, Romania, by sequencing the p72 gene. The results obtained show that all Romanian samples have 100% identity with all compared isolates from Georgia, Armenia, Russia, Azerbaijan, Ukraine, Belarus, Lithuania, and Poland.

Since the manuscript reports for the first time the existence of the ASFV genotype II in the country, the results have some interest. However, the authors must perform the following changes, before the acceptance of the their work:

1 - The sentence "The swine populations that are positive to ASF are currently eradicated..." is not correct

2 - Also in page 2, lines 75-77, don´t report the truth of the history. The genotype I was firstly identified in Portugal and Spain.

3 - In Introduction, page 2, line 93, authors should add some features about the viral DNA replication in the cell, after the sentence "...virus with an icosahedral morphology and a double-stranded DNA [12]."

4 -In M&M, results and discussion sections, authors should compare the obtained p72 sequences with the new ones reported in Chinese isolates. This request is particularly relevant to improve the final quality of the work

5 - In the results section, page 4, line 160, the authors stated that "The presence of ASFV was confirmed in all nine tested samples". I think that is not correct, because authors only analyzed the presence of viral DNA.

6 - Finally, all the manuscript should be review by a native English speaker.

Author Response

The manuscript submitted by Ungur et al., entitled "Genotyping of African swine fever virus (ASFV) isolates in Romania with the first report of genotype II in symptomatic pigs" aims to characterize nine African swine fever virus (ASFV) isolates obtained from domestic pigs in MureÅŸ County, Romania, by sequencing the p72 gene. The results obtained show that all Romanian samples have 100% identity with all compared isolates from Georgia, Armenia, Russia, Azerbaijan, Ukraine, Belarus, Lithuania, and Poland.

Since the manuscript reports for the first time the existence of the ASFV genotype II in the country, the results have some interest. However, the authors must perform the following changes, before the acceptance of their work:

Re: The authors thank Reviewer 1 for the revision of the submitted manuscript. A point-by-point response is given below for each comment/suggestion:

1 - The sentence "The swine populations that are positive to ASF are currently eradicated..." is not correct

Re: Lines 41-43. The authors rephrased the sentence to ‘The swine populations that are positive to ASF are currently stunned, and the carcasses destroyed by incineration.  In some circumstances, a prophylactic depopulation is also performed throughout the positive regions.’

2 - Also in page 2, lines 75-77, don´t report the truth of the history. The genotype I was firstly identified in Portugal and Spain.

Re: Lines 78-80. The authors changed the mentioned phrase to “The genotype I incursion started in Portugal in 1960 with spread to Spain and several other European countries, and it was eradicated by 1995 with the exception of Sardinia, Italy where it remained endemic.”

3 - In Introduction, page 2, line 93, authors should add some features about the viral DNA replication in the cell, after the sentence "...virus with an icosahedral morphology and a double-stranded DNA [12]."

Re: Lines 95-99.The authors included the following: “The main site of replication is the cytoplasm of the infected cells, although an earlier stage of nuclear DNA synthesis has been documented, DNA replication occurs in the peri-nuclear factory site in about 6 hours after infection. Enzymes essential for virus genome transcription and replication, as well as virion structural proteins, are encoded in the virus genome”

4 -In M&M, results and discussion sections, authors should compare the obtained p72 sequences with the new ones reported in Chinese isolates. This request is particularly relevant to improve the final quality of the work

Re: The authors thank the reviewer for the suggestion. The lines 192-195 were added for a better understanding.

5 - In the results section, page 4, line 160, the authors stated that "The presence of ASFV was confirmed in all nine tested samples". I think that is not correct, because authors only analyzed the presence of viral DNA.

Re: Lines 166-168.The sentence was corrected to ‘The presence of the viral DNA was confirmed in all nine tested samples. All positive samples were successfully sequenced and phylogenetically analyzed for p72 (B646L) targeted gene.’

6 - Finally, all the manuscript should be review by a native English speaker.

Re: The authors thank Reviewer 1 for all the comments and recommendations. Extensive editing has been made to the paper.

Reviewer 2 Report

Information on the phylogeny of outbreak viruses is always welcome, adds to the growing store of knowledge and confirms much of what we already know about ASF virus. I believe this information should be published, but the manuscript needs a fair amount of attention before it can be accepted for publication.

As detailed below, the Introduction is poorly researched and contains a good deal of totally incorrect statements that need to be removed. In a paper like this a much shorter introduction, with just the basic information about the general distribution of the epidemic and focusing on Romania would be acceptable, but if the longer version is preferred the authors must correct all the misinformation contained in the current Introduction.

The English requires attention, I have not provided extensive editing (which needs to be applied) but some of the glaring errors that need to be fixed are listed under Minor revisions.

I detected some misspelling of author names in the reference list, I did not go through the list exhaustively but that is careless and in fact offensive and the list must be gone through very carefully to make sure that all of the authors’ names in the reference list are correct and everything else about the references is correct too.

Major revision needed

Line 51: Worldwide is prophetic rather than actual at the moment. Rather use widespread or be more specific – it is widespread in Africa, Asia and Europe. It has furthermore so far affected one island nation in Oceania and one island that is home to two nations in the Caribbean region of the Americas. As the last two recent incursions are not very relevant for this paper, I think ‘widespread in Africa, Asia and Europe would do very well and better describe the situation than worldwide, even if that might be true by this time next year!

Lines 51-52: Recent literature from Europe and Asia has confirmed that ASF is not ‘highly’ contagious, in fact it spreads rather slowly compared with other diseases, e.g. classical swine fever and foot and mouth disease (see Busch et al., 2021 attached, it has some diagrams that explain this very well).

Lines 74-77: This information is incorrect (severely so). The genotype I incursion that eventually affected Sardinia started in Portugal in 1960 with spread to Spain, several other countries in western Europe (Netherlands and Belgium are hardly southern or even central) as well as two islands in the Caribbean and Brazil. It reached Sardinia in 1978 and became endemic there, leaving it as the only area outside Africa to be infected after the disease was finally eradicated from Portugal and Spain by 1995. It never spread anywhere else from Sardinia so it is quite untrue that the cluster started there or that it spread from there. As this highly inaccurate statement is not referenced, I suggest reading some of the literature, providing correct information and referencing it. A review that describes the events well is attached (Costard et al., 2009).

Line 79: It is not at all likely that Asia became infected by spread from Georgia, which hasn’t reported an outbreak since 2011. Molecular genetic analysis of the first genotype II virus to be isolated in China showed it to be most closely related to strains circulating in Russia and other eastern European countries (Poland and Estonia) (Bao et al., 2019).

Lines 80-81: ASF has definitely not spread to ALL European Union countries – Austria, Croatia, Cyprus,  Denmark, Finland, France, Ireland, Italy, Luxembourg, Malta, Netherlands, Portugal, Slovenia, Spain, and Sweden are all EU member countries that have NOT reported ASF (more than half of the EU member states).

Line 103: Omit ‘widely distributed’ – all but genotypes I and II are restricted to southern and eastern Africa and none are found throughout the entire region, some have only been found at one or two limited locations.

Lines 197-198: The statement exaggerates what phylogenetic studies are able to do. They can certainly demonstrate the relatedness amongst outbreak viruses, but they alone cannot determine the origin or even the direction of transmission without some fire engine history, e.g. importation of pigs or pork fed as swill from a specific place during a specific period of time. As the authors indicate, the genotype II viruses in Asia and Europe dating from the 2007 introduction in Georgia are now so widespread and so genetically similar that p72 sequencing is uninformative epidemiologically and biologically. However, sub-genotypic studies based on variable regions like CVR can provide more information (Gallardo et al., 2014). However, even at this higher resolution it is not possible to link an outbreak virus with a specific source without detailed history obtained on farm (Goller et al., 2015; Kolbasov et al. 2018)

Minor corrections needed

Line 41: Replace ‘confine’ with ‘quarantine’. The meaning of confine is unclear in this context and one cannot confine a region.

Line 67: Replace ‘hearth’ with ‘heart’ (the former means fireplace).

Line 81: Replace In with An (beginning of sentence).

Line 90: Replace ‘growth’ with ‘husbandry’ or ‘rearing’.

Line 91: Replace ‘hosting’ with ‘housing’.

Line 185: The use of ‘neither’ is incorrect here, as it can only be used when two items are being compared, whereas I presume this refers to the multiple virus sequences, i.e. none of them differed genetically from the others.

Line 225: The first author’s name is Njau (not Nijau).

Line 245: Is this reference compliant with the journal’s guidelines?

Line 256: Replace Kolvasov with Kolbasov.

Author Response

Information on the phylogeny of outbreak viruses is always welcome, adds to the growing store of knowledge and confirms much of what we already know about ASF virus. I believe this information should be published, but the manuscript needs a fair amount of attention before it can be accepted for publication.

Re: The authors thank Reviewer 2 for the meticulous revision of the submitted manuscript. A point-by-point response is given below for each comment/suggestion.

As detailed below, the Introduction is poorly researched and contains a good deal of totally incorrect statements that need to be removed. In a paper like this a much shorter introduction, with just the basic information about the general distribution of the epidemic and focusing on Romania would be acceptable, but if the longer version is preferred the authors must correct all the misinformation contained in the current Introduction.

Re: The authors initially have written this paper as a short communication, and it was previously submitted in a much shorter version, but it was rejected because the minimum word count (2000 words) and number of references (20) were not met.

The English requires attention, I have not provided extensive editing (which needs to be applied) but some of the glaring errors that need to be fixed are listed under Minor revisions.

Re: The authors thank Reviewer 2 for all the recommendations, which were accepted by the authors. The English language has been extensively revised.

I detected some misspelling of author names in the reference list, I did not go through the list exhaustively but that is careless and in fact offensive and the list must be gone through very carefully to make sure that all of the authors’ names in the reference list are correct and everything else about the references is correct too.

Re: The comments are much appreciated. The authors corrected the names is the reference list.

Major revision needed

Line 51: Worldwide is prophetic rather than actual at the moment. Rather use widespread or be more specific – it is widespread in Africa, Asia and Europe. It has furthermore so far affected one island nation in Oceania and one island that is home to two nations in the Caribbean region of the Americas. As the last two recent incursions are not very relevant for this paper, I think ‘widespread in Africa, Asia and Europe would do very well and better describe the situation than worldwide, even if that might be true by this time next year!

Re: Line 50. The authors thank Reviewer 2 for the comments, and they changed all the sentences that intuited the global spread of the ASFV

Lines 51-52: Recent literature from Europe and Asia has confirmed that ASF is not ‘highly’ contagious, in fact it spreads rather slowly compared with other diseases, e.g. classical swine fever and foot and mouth disease (see Busch et al., 2021 attached, it has some diagrams that explain this very well).

RE: Lines 51-53. The authors added the new information and rephrased the sentence to “ASF is a viral disease widespread in Africa, Asia and Europe, with acute-to-chronic manifestation and a hemorrhagic character. Recent studies have infirmed the highly contagious character, and proved that ASF spreads rather slowly compared with other infectious diseases.”

Lines 74-77: This information is incorrect (severely so). The genotype I incursion that eventually affected Sardinia started in Portugal in 1960 with spread to Spain, several other countries in western Europe (Netherlands and Belgium are hardly southern or even central) as well as two islands in the Caribbean and Brazil. It reached Sardinia in 1978 and became endemic there, leaving it as the only area outside Africa to be infected after the disease was finally eradicated from Portugal and Spain by 1995. It never spread anywhere else from Sardinia so it is quite untrue that the cluster started there or that it spread from there. As this highly inaccurate statement is not referenced, I suggest reading some of the literature, providing correct information and referencing it. A review that describes the events well is attached (Costard et al., 2009).

Re: Lines 76-78. The authors changed the mentioned phrase to “The genotype I incursion started in Portugal in 1960 with spread to Spain and several other European countries, and it was eradicated by 1995 with the exception of Sardinia, Italy where it remained endemic.”

Line 79: It is not at all likely that Asia became infected by spread from Georgia, which hasn’t reported an outbreak since 2011. Molecular genetic analysis of the first genotype II virus to be isolated in China showed it to be most closely related to strains circulating in Russia and other eastern European countries (Poland and Estonia) (Bao et al., 2019).

Re: Lines 80-81. The authors changed the phrase to: “A recent study confirmed that the circulating strains of ASFV in Asia are derived from the ones in Europe and Russia”

Lines 80-81: ASF has definitely not spread to ALL European Union countries – Austria, Croatia, Cyprus, Denmark, Finland, France, Ireland, Italy, Luxembourg, Malta, Netherlands, Portugal, Slovenia, Spain, and Sweden are all EU member countries that have NOT reported ASF (more than half of the EU member states).

Re: Lines 82-83. The authors rephrased the sentence to “ASF was first reported in the European Union in Lithuania in early 2014, and it has since spread to several European Union countries, including Romania.”

Line 103: Omit ‘widely distributed’ – all but genotypes I and II are restricted to southern and eastern Africa and none are found throughout the entire region, some have only been found at one or two limited locations.

Re: Line 110. The authors changed “widely distributed” with “reported in different geographical regions”

Lines 197-198: The statement exaggerates what phylogenetic studies are able to do. They can certainly demonstrate the relatedness amongst outbreak viruses, but they alone cannot determine the origin or even the direction of transmission without some fire engine history, e.g. importation of pigs or pork fed as swill from a specific place during a specific period of time. As the authors indicate, the genotype II viruses in Asia and Europe dating from the 2007 introduction in Georgia are now so widespread and so genetically similar that p72 sequencing is uninformative epidemiologically and biologically. However, sub-genotypic studies based on variable regions like CVR can provide more information (Gallardo et al., 2014). However, even at this higher resolution it is not possible to link an outbreak virus with a specific source without detailed history obtained on farm (Goller et al., 2015; Kolbasov et al. 2018)

Re: Lines 214-215. The authors removed the phrase “is critical for determining the pathogen’s origin and transmission chains”.

Minor corrections needed

Line 41: Replace ‘confine’ with ‘quarantine’. The meaning of confine is unclear in this context and one cannot confine a region.

Re: Line 40. Replaced.

Line 67: Replace ‘hearth’ with ‘heart’ (the former means fireplace).

Re: Line 67. Corrected.

Line 81: Replace In with An (beginning of sentence).

Re: Line 83. Replaced.

Line 90: Replace ‘growth’ with ‘husbandry’ or ‘rearing’.

Re: Line 92. Replaced.

Line 91: Replace ‘hosting’ with ‘housing’.

Re: Line 93. Replaced.

Line 185: The use of ‘neither’ is incorrect here, as it can only be used when two items are being compared, whereas I presume this refers to the multiple virus sequences, i.e. none of them differed genetically from the others.

Re: Line 197. Corrected.

Line 225: The first author’s name is Njau (not Nijau).

Re: Line 241. Replaced.

Line 245: Is this reference compliant with the journal’s guidelines?

Re: The authors think that the reference is compliant with the journal’s guidelines. A copy-pasted instruction from https://www.mdpi.com/authors/layout#_bookmark53 is given below. We did not find instructions exactly for an official institution such as World Organisation for Animal Health, but we adapted the one below from a company website:

‘Company website

Proto Labs Ltd. Protolabs. Available online: https://uploads.protolabs.co.uk/ es/PartUpload-MultiPart.aspx?LinkFrom=FC (accessed on 24 April 2017).’

Line 256: Replace Kolvasov with Kolbasov.

Re: Line 283. Replaced.

Round 2

Reviewer 1 Report

The authors answered positively to the majority of the questions raised by this reviewer. When the stated "Re: Lines 95-99.The authors included the following: “The main site of replication is the cytoplasm of the infected cells, although an earlier stage of nuclear DNA synthesis has been documented, DNA replication occurs in the peri-nuclear factory site in about 6 hours after infection. Enzymes essential for virus genome transcription and replication, as well as virion structural proteins, are encoded in the virus genome”, they should add the following references:

https://doi.org/10.3390/v7092858 

https://doi.org/10.1016/j.virusres.2015.07.006

Author Response

Reviewer 1

The authors answered positively to the majority of the questions raised by this reviewer. When the stated "Re: Lines 95-99. The authors included the following: “The main site of replication is the cytoplasm of the infected cells, although an earlier stage of nuclear DNA synthesis has been documented, DNA replication occurs in the peri-nuclear factory site in about 6 hours after infection. Enzymes essential for virus genome transcription and replication, as well as virion structural proteins, are encoded in the virus genome”, they should add the following references:

https://doi.org/10.3390/v7092858 

https://doi.org/10.1016/j.virusres.2015.07.006

Re: The authors thank Reviewer 1 for the recommendations. The references were added in line 97.

Reviewer 2 Report

The manuscript has improved substantially and most of the reviewer concerns have been adequately addressed. Two further changes are required:

Line 52: I am not sure what ‘infirmed’ the highly contagious character of the disease means, but what the article cited demonstrates is that ASF should not be described as ‘highly contagious’ because compared with diseases like FMD and CSF it spreads slowly due to low contagiousness. Replace ‘infirmed’ (no such word exists in English, although someone ill or weak s said to be infirm) with contradicted or disputed.

Line 115: Replace ‘reported in different geographical regions’ with ‘reported in eastern and southern Africa, with genotypes I and II having become established in other regions as well’, which is true. The statement as it stands is rubbish, as ‘different geographical regions’ to most readers would imply regions other than Africa, let alone a limited part of Africa. I did explain this in the first review round where the authors had stated ‘widely distributed’ so I am not sure why it was changed to something equally inaccurate. This is what I asked for before: ‘Line 103: Omit ‘widely distributed’ – all but genotypes I and II are restricted to southern and eastern Africa and none are found throughout the entire region, some have only been found at one or two limited locations.’ I guess I over-explained! Please replace text as indicated above. 

Author Response

Reviewer 2

The manuscript has improved substantially and most of the reviewer concerns have been adequately addressed. Two further changes are required:

Line 52: I am not sure what ‘infirmed’ the highly contagious character of the disease means, but what the article cited demonstrates is that ASF should not be described as ‘highly contagious’ because compared with diseases like FMD and CSF it spreads slowly due to low contagiousness. Replace ‘infirmed’ (no such word exists in English, although someone ill or weak s said to be infirm) with contradicted or disputed.

Re: Line 52. The authors replaced 'infirmed' with 'disputed'.

Line 115: Replace ‘reported in different geographical regions’ with ‘reported in eastern and southern Africa, with genotypes I and II having become established in other regions as well’, which is true. The statement as it stands is rubbish, as ‘different geographical regions’ to most readers would imply regions other than Africa, let alone a limited part of Africa. I did explain this in the first review round where the authors had stated ‘widely distributed’ so I am not sure why it was changed to something equally inaccurate. This is what I asked for before: ‘Line 103: Omit ‘widely distributed’ – all but genotypes I and II are restricted to southern and eastern Africa and none are found throughout the entire region, some have only been found at one or two limited locations.’ I guess I over-explained! Please replace text as indicated above.

Re: Lines 115-116. The authors thank Reviewer 2 for the clarification. They replaced 'reported in different geographical regions' with 'reported in eastern and southern Africa, with genotypes I and II having become established in other regions as well'.

This manuscript is a resubmission of an earlier submission. The following is a list of the peer review reports and author responses from that submission.